# Occurrence of Nine Pyrrolizidine Alkaloids in *Senecio vulgaris* L. Depending on Developmental Stage and Season

**DOI:** 10.3390/plants8030054

**Published:** 2019-03-05

**Authors:** Jens Flade, Heidrun Beschow, Monika Wensch-Dorendorf, Andreas Plescher, Wim Wätjen

**Affiliations:** 1Plant Nutrition, Institute of Agricultural and Nutritional Sciences, Martin-Luther-University Halle-Wittenberg, Betty-Heimann-Strasse 3, 06120 Halle/Saale, Germany; Jens.Flade@gmx.de (J.F.); heidrun.beschow@landw.uni-halle.de (H.B.); 2PHARMAPLANT Arznei- und Gewürzpflanzen Forschungs- und Saatzucht GmbH, Am Westbahnhof 4, 06556 Artern, Germany; info@pharmaplant.de; 3Biofunctionality of Secondary Plant Compounds, Institute of Agricultural and Nutritional Sciences, Martin-Luther-University Halle-Wittenberg, Weinbergweg 22, 06120 Halle/Saale, Germany; 4AG Biometrie und Agrarinformatik, Institute of Agricultural and Nutritional Sciences, Martin-Luther-University Halle-Wittenberg, Karl-Freiherr-von-Fritsch-Strasse 4, 06120 Halle/Saale, Germany; monika.dorendorf@landw.uni-halle.de

**Keywords:** pyrrolizidine alkaloids, *Senecio vulgaris* L., ontogenetically variation, toxic plant compounds

## Abstract

The contamination of phytopharmaceuticals and herbal teas with toxic plants is an increasing problem. *Senecio vulgaris* L. is a particularly noxious weed in agricultural and horticultural crops due to its content of toxic pyrrolizidine alkaloids (PAs). Since some of these compounds are carcinogenic, the distribution of this plant should be monitored. The amount of PAs in *S. vulgaris* is affected by various factors. Therefore, we investigated the occurrence of PAs depending on the developmental stage and season. A systematic study using field-plot experiments (four seasons, five developmental stages of the plants: S1 to S5) was performed and the PA concentration was determined via LC-MS/MS analysis. The total amount of PAs in the plant increased with the plant development, however, the total PA concentrations in µg/g dry matter remained nearly unchanged, whilst trends for specific PAs were observed. The concentrations of PA-*N*-oxides (PANOs) were much higher than that of tertiary PAs. Maximal amounts of the PA total were 54.16 ± 4.38 mg/plant (spring, S5). The total amount of PAs increased strongly until later developmental stages. Therefore, even small numbers of *S. vulgaris* may become sufficient for relevant contaminations set out by the maximal permitted daily intake levels recommended by the European Food Safety Authority (EFSA).

## 1. Introduction

Pyrrolizidine alkaloids (PAs) are phytochemicals that are thought to be occurring in more than 6000 plant species [1,2], e.g., Asteraceae, Boraginaceae, Fabaceae and Orchidaceae families [3,4]. Until recently, 700 plant species are indicated to be able to form PAs [5]. Currently, more than 500 different PAs are known [6]. Various PAs can cause hepatotoxicity (veno-occlusive disease) in animals and humans. Particularly, 1,2-unsaturated PAs are of great concern [6,7]. It has been shown that specific PAs could also act as genotoxic carcinogens, causing cancer in rodents [7,8]. In consequence of their toxic properties, the evaluation of PAs as contaminants in plant-derived medicinal and food products has come more and more into the focus of scientific research [7,9,10,11,12].

A broadening in the geographic distribution of PA-containing plants has been observed in the last several years. This finding correlates with an increasing concentration of PAs in harvested plants cultivated for the production of food or herbal medicines [1,12,13,14,15]. It is well known that *Senecio vulgaris* L. (common groundsel), belonging to the Asteraceae family, is able to synthesize PAs [5,16,17,18,19]. Because of the very high potential of the plant for reproduction and spreading [14], the occurrence of PA producing *S. vulgaris* in agricultural, as well as horticultural plant cultures, is extremely problematic [7,15,20]. During the last years, contaminations by parts of this plant were detected in harvested crops, especially in cultures of morphologically similar plants like *Diplotaxis tenuifolia* L. and *Eruca sativa* L. The risks of such an accidental intake of PAs via contaminated plant products were already intensively discussed [13,15,20,21].

Various factors can contribute to qualitative [22,23,24,25,26,27,28,29,30,31] as well as quantitative [15,22,32,33,34,35,36,37,38,39,40,41,42,43,44,45,46] parameters of the PA synthesis/metabolism in *Senecio* species, leading to diverse overall PA spectra in these plant species [47]. On grounds of the wide occurrence of *S. vulgaris* in economically used plant cultures [7,14,15,48,49], qualitative as well as quantitative changes in the PA spectrum of this plant lead to different contamination scenarios, that may have a great impact on potential risks for humans and animals [36,46]. Furthermore, it is known that PA contaminations are often the reason for spot contamination in distinct charge numbers [50,51,52]. It cannot be excluded that this disbalance in contamination load may be caused by differences in the PA content of single *S. vulgaris* plants. Due to the high variability of PAs in different plants (PA concentration, the relationship of tertiary PAs vs. PA-*N*-oxides, the occurrence of distinct PA structures), a general estimation of the toxicological risk of PA-containing plants for humans is rather problematic [7].

At the moment, there are no legal limit values for 1,2-unsaturated PAs in foods and feeds. Within the European Union, the general recommendation applies that exposure to genotoxic and carcinogenic substances should be minimized to the lowest level achievable by reasonable means (ALARA principle: As low as reasonably achievable) [50,51,52]. From official surveys, experimental work and reviews carried out or compiled for herbal products and honey, it becomes clear that a wide range of matrices consumed can be contaminated with PAs: (i) Honey and pollen [11,21,53,54,55,56,57,58,59], (ii) herbal medicines [51,60,61], (iii) herbal supplements and foods [9,11,13,15,20,59] and (iiii) tea and herbal teas [60,61,62,63,64,65,66]. According to calculations of the German Federal Institute for Risk Assessment (BfR), the intake of 1,2-unsaturated PAs by children aged six months to five years is mainly attributable to herbal teas (including rooibos tea), black tea and honey. With adults, the contribution to total 1,2-unsaturated PA intake made by honey is lower, and that of green tea higher, than with children [10,50]. As a consequence, contamination of herbal products with *S. vulgaris* may be a relevant source of PAs.

We used HPLC-MS analysis to clarify if the concentration and amount of the total PAs in *S. vulgaris* is influenced by ontogenesis and season. Furthermore, we investigated the influence of ontogenesis and season on the nine distinct PAs, which were detected in this plant (Figure 1). The present study also aimed to elucidate in which developmental stages *S. vulgaris* has its highest PA concentrations and amounts, and to clarify whether there is a change of PA concentration during ontogenesis. Finally, a conclusion was objected which PAs in *S. vulgaris* represent the main alkaloids and whether they are differently ontogenetically influenced in different seasons. This was done by performing a systematic study using field-plot experiments (Appendix A). The results of this study are important for the risk management of plants for nutritional purposes, which are contaminated by *S. vulgaris*.

## 2. Results

In four seasons (spring, early summer, midsummer, autumn) nine PAs were investigated in five developmental stages (S1 to S5, according to Table 2). The data for the PA concentration and amount were presented as the sum of all nine PAs (“total”), and also subdivided in the sum of the five tertiary PAs (“tertiary”: retrorsine, Re; seneciphylline, Sp; senecivernine, Sv; senecionine, Sc; and senkirkine; Sk) and the sum of the four PA-*N*-oxides (“PANO”: retrorsine-*N*-oxide, ReN; seneciphylline-*N*-oxide, SpN; senecivernine-*N*-oxide, SvN; and senecionine-*N*-oxide, ScN). 

First, we analyzed the PA concentration in μg/g dry matter (dm) in *S. vulgaris*. It could be shown that these sums were different in the developmental stages, to the respective individual seasons (Figure 2). During the year, the total PA concentrations in *S. vulgaris* showed a year-round PA abundance of at least 1654.3 ± 394.1 μg/g (S5, autumn) and a maximum of 4910.2 ± 1349 μg/g (S2, midsummer). There were a few significant exceptions to the similar magnitude of the total PA concentrations during ontogenesis: (i) The total PAs in spring ranged from 2610.8 ± 581.2 µg/g (S1) to 3573.8 ± 243.2 µg/g (S3). This discrepancy was significant, but probably not relevant for biological systems; (ii) In early summer, there were also significantly elevated concentrations of total PA from S1 (2013.5 ± 254.3 μg/g) to S3 (3275.4 ± 240.1 μg/g) and S4 (3963.1 ± 1216.2 μg/g); (iii) while in autumn, the highest concentrations were detected in S1 (4072.5 ± 846.1 μg/g). The PA concentrations (“total” and “PANO”) of midsummer did not show any statistically significant difference. Consistently in all developmental stages and in each season, the concentrations of the PA-*N*-oxides (PANOs) were higher than the tertiary PAs in the plant, leading to ratios of tertiary PAs to PANOs between 10:90 to 30:70. The minimal difference between PANOs and tertiary PAs was 2.7-fold (spring, S2) and the maximal difference was fourteen-fold (early summer, S5). For the total PA concentration (µg/g) in the fresh weight (fw) see Appendix A.

Secondly, we determined the PA amount in mg/plant of *S. vulgaris* depending on the developmental stage (S1 to S5) and season (Figure 3). The calculation of the PA amount (mg/plant) was based on the PA concentration (µg/g dm) and dry matter per plant (for the values of the dry matter, see Appendix A). In all seasons, the PA amount per plant to full flowering (S4) was decisively influenced by the growth of the plant (in most cases: Increase in height, dry matter and fresh weight; Appendix A). In full flowering (S4) and fruit development and seed maturity (S5), a significant difference in all seasons was observed. Particularly noteworthy were the high total PA amounts in spring and early summer, whereas significantly reduced values were present in midsummer and autumn. In midsummer, the average amount of total PAs in S4 was only 26–38% and in S5 only 17–24% compared to the previous seasons. The PA levels fell in the autumn in S4 and S5, compared to spring (about 29.7% and 69.5%) and early summer (about 52.6% and 56.8%). In spring, early summer and midsummer the magnitude of the total amount of PA per plant of S1, S2 and S3 were each about likewise. The highest amount of total PAs was found in spring, S5, with a value of 54.16 ± 4.38 mg/plant. In comparison, at the same developmental stage in midsummer, the total PA amount was 5.8-fold lower (9.33 ± 2.67 mg/plant).

Thirdly, we investigated the PA concentration of the nine PAs (five tertiary PAs, four PANOs) separately (Figure 4). Until midsummer, the tertiary PAs were more likely to be influenced than the PANOs, whereas in autumn, only the tertiary PAs senecionine and senkirkine were affected. Therefore, changes in the concentrations of the *N*-oxides prevailed at this time of the year. 

Distinct changes in single PA concentrations were observed depending on the developmental stage and season (Figure 4, Appendix A). Sk was the alkaloid that was detected only in traces below 1 µg/g and was therefore much lower than the other PAs investigated, e.g., with the highest value: 0.28 µg/g (midsummer, S5). In the case of the most abundant alkaloids (ReN, SpN, ScN), the changes of these three alkaloids were described pairwise (tertiary PAs and corresponding PANOs).

Re was detected only in very small concentrations, ranging from 2.8 µg/g (spring, S1) to 61.1 µg/g (midsummer, S4). Regarding the corresponding *N*-oxide (ReN) in midsummer, S4; the concentration of ReN was approximately eighteen-fold higher (1099.3 µg/g) than that of Re. In all seasons (except autumn), we found a strong increase in the Re/ReN concentrations during ontogenesis, e.g., the ReN concentration in spring: 15.2 µg/g (S1) to 739.0 µg/g (S5). In autumn, however, only relatively low concentrations of Re/ReN were detectable. 

Furthermore, the corresponding alkaloids Sp/SpN were investigated and a higher concentration of the *N*-oxide (SpN) was found compared to the tertiary PA (Sp). These corresponding PAs accumulated in high concentrations particularly in the early developmental stages S1 and S2, with initial concentrations of SpN varying between 664.8 µg/g (early summer) and 1154.8 µg/g (spring). Therefore, in contrast to the ontogeny increase of ReN, a nearly inverse trend was observed for SpN. A significant decrease in the concentration of SpN was recognizable during ontogenesis, e.g., in spring: Reduction to 44.5% of the initial value (1154.8 µg/g to 514.5 µg/g). 

Regarding Sc/ScN, high PA concentrations appeared in the earliest developmental stage (highest S1 value in autumn: 2297.2 µg/g). In contrast to Re/ReN and Sp/SpN, no general trend was observed in the concentrations of ScN: In early summer, the concentration increased nearly 2.4 -fold from S1 to S4, while in autumn, a decrease of more than 60% was recognizable between S1 and S2.

## 3. Discussion

Contamination of herbal medicines and plant-derived foods with toxic weeds of the *Senecio* species (e.g., *S. vulgaris*) leads to an increasing problem due to the weeds’ rapid growth. These plants are of high risk for consumers, depending on their specific spectrum of toxic PAs. In animal studies, certain PAs found in *S. vulgaris* show hepatotoxic, as well as carcinogenic and genotoxic effects [7,8]. This applies to PAs with a 1,2-unsaturated necine base that is further esterified with at least one branched C_5_-carboxylic acid [1,10,53,60]. The corresponding PA-*N*-oxides are less toxic [6,7]. Our study has shown that senecionine-*N*-oxide (ScN), seneciphylline-*N* oxide (SpN) and retrorsine-*N*-oxide (ReN) were the dominating PAs in *S. vulgaris* during ontogeny in different seasons. In all developmental stages and seasons, the total concentrations of PAs (µg/g dm) in *S. vulgaris* remained nearly constant and the proportion of PANOs was much higher than that of tertiary PAs. In contrast to that, changes in distinct PAs were significant for the developmental stage and highly influenced by the season. 

The annual mean during full flowering (S4) of the total PA amounts in *S. vulgaris* of our present study (annual mean values of total PA concentration and amount not shown) are comparable to values reported by Hartmann and Zimmer [38]: The PA amount in flowering plants (S4) was reported to be 25.9 mg/plant, which is analogous to our value (23.4 mg/plant). In virtues of the differences in the average fresh weight (57.3 g and 49.4 g per plant, respectively), the PA concentrations based on fresh weight were somewhat lower (465.2 µg/g compared to 524 µg/g). Regarding developmental stage S2, this was characterized with 3 to 6 sprouts per plant in the present study, while Hartmann and Zimmer have sampled plants with 10 to 15 flowers (plant height 15 cm). Thus, the results in the two studies differ: While the fresh weight for this stage was comparable (8.4 g and 9.1 g, respectively), the PA concentration in our study was about three times higher (362.4 µg/g fw compared to 129 µg/g fw). This discrepancy might demonstrate the effect of further factors regulating the PA content in the plants, like nutrient and water supply or environmental stresses. 

The results from early summer in our study are representative for the average of the four studied seasons of the year, with a slow increase in total PA concentration in the dry matter until the developmental stage S4 (full flowering) and a tendency for the decrease at the end of the life cycle (S5). Similar results for an increase in total PA concentration until full flowering have been shown before for the PA concentration, based on fresh weight in *S. vulgaris* by Hartmann and Zimmer [38] and for other *Senecio* species by Berendonk et al. [22]. Furthermore, a continuous increase of PAs in seedlings of *S. vulgaris* up to 30 days after germination was demonstrated by Schaffner et al. [45]. In the other seasons, the ontogenetic differences in total PA concentration in the present study are not pronounced (midsummer), less pronounced (spring) or different in patterns (autumn).

In our study, the increased total PA amount in plants of *S. vulgaris* during their development correlated with an increase in biomass at nearly constant PA concentrations. That has its cause in the conjunction of the growth of aboveground biomass (stems, leaves, flowers) with increasing biomass of the roots, where the PA synthesis in *S. vulgaris* is located [39,45]. Since the PA concentration is only slightly compensated with increasing growth, it is possible that when an increase in biomass occurs quickly, the PA concentration may decrease due to relatively slower PA synthesis [39,67]. We observed such a slight decrease in the total PA concentration at full flowering in spring, contrary to the above-mentioned tendency of a slight increase in total PA concentration until full flowering.

In the autumn, a highly significant total concentration decline from S1 (4072.5 ± 846.1 μg/g) to S2 (2165.7 ± 269.3 μg/g) was observed. Also, other authors were able to determine such a concentration change in *S. vulgaris* by a factor of around two, compared to the previous sampling date [37]. Compared to the other seasons, in our study, the plants in S1 showed in autumn the lowest dry matter, which means that the synthesized PAs in the plants may have been concentrated at this developmental stage.

During the development of *S. vulgaris*, dynamic changes in the profile of specific PAs have occurred. The concentration of distinct PAs has significantly changed during development (e.g., retrorsine-*N*-oxide, senecionine), while other PAs (e.g., senecivernine-*N*-oxide) remained nearly unchanged. The fact that significant changes in concentrations of senecivernine-*N*-oxide only occurred during midsummer may be since this PA is synthesized together with senecionine-*N*-oxide as a primary product in the roots of *S. vulgaris* [26,47]. It is conceivable that senecivernine-*N*-oxide converts very quickly to other PAs, or that the small amounts formed do not give rise to further PAs. At the same time, it is noticeable that in midsummer, senecionine-*N*-oxide showed no changes during ontogenesis. Perhaps this is an indication that under certain physiological conditions, such as drought and high temperatures in midsummer, senecivernine-*N*-oxide is more likely to undergo concentration changes. Regarding the significant concentration changes from S3 to S5 in spring, early summer and midsummer; between both development stages the ontogenetical increase of ReN and the decrease of SpN, may go in line with the conclusions from in vitro experiments by Hartmann and Dierich [26]. The authors stated that SpN was modified very quickly from ScN after this was synthesized. Furthermore, ReN occurred late in the experiments. Therefore, we assumed by virtue of our present results, that ReN and SpN may undergo an interchangeable transformation during ontogenesis in *S. vulgaris*.

*S. vulgaris* qualitatively displays a similar PA profile as *S*. *vernalis* [68] and our studies have confirmed that the quality of the PA profile of *S. vulgaris* differs markedly from many other species of the Senecioneae tribe during ontogenesis [22,23,24,27,28,30,31,33,36,42,43,46,69,70,71]. Respective to the concentrations of specific PAs in *S. vulgaris*, our study has clearly shown that they were dependent on the growth and the developmental stage of the plant. This result could be used for qualitative/quantitative estimations of the PA spectrum of *S. vulgaris* and thus for a more realistic risk management of this plant as (i) a contaminant of medical plants and (ii) plant-derived food, considering the different toxic potential of the different PAs [10,72].

## 4. Materials and Methods 

The study (experiments in open land, sample preparation for LC-MS/MS-analysis) was conducted at PHARMAPLANT Arznei- und Gewürzpflanzen Forschungs- und Saatzucht GmbH (Am Westbahnhof 4, 06556 Artern, Germany), the field experiments were conducted during a time period from 30 March, 2016 until 3 December, 2016.

The *S. vulgaris* plants used in this study were obtained from seeds collected in 2015 (PHARMAPLANT), and germinated and cultured to ready-to-plant plants in a temperate greenhouse. The temperatures in the greenhouse fluctuated throughout the seasons, but were never above 32 °C or below 13 °C. Approximately one week before the trials in open land were established, the plants were transferred outside the greenhouse for final cultivation and for adaption to the current weather conditions before planting.

The study was conducted in four sub-trials on the basis of four different seasons being investigated: Spring (start: 30 March, 2016), early summer (start: 20 May, 2016), midsummer (start: 7 July, 2016) and autumn (start: 2 September, 2016). The sampling took place on the respective days at 13:00. For the single dates for sampling, see Table 1. 

Relevant for the development of the plants (ontogenesis), five development stages were defined (S1 to S5, Table 2) according to the BBCH scale for weeds listed in the monograph of Meier [73]. For each of these five developmental stages (S1 to S5) five independent field plots were prepared for each of the four seasons (see Appendix A). Each field plot had a size of 5.54 m^2^ (2.8 × 1.98 m) with a plant density of 15.15 plants/m^2^. At sampling, 40 single plants were collected from each field plot. The experimental design within each season corresponds to a Latin square for the effect stage (Appendix A).

The plant material (all aboveground parts) was dried (60 °C) to a constant weight and subsequently crushed with a cutting mill (4 mm) before transporting the samples to the laboratory (Institut Kirchhoff Oudenarder Straße 16/Carrée Seestraße, 13347 Berlin-Mitte, Germany) for the analysis of the PAs. The material was further ground (0.25 mm) and finally dried (4 h, 103 °C) for the determination of dry mass. The analysis of the PAs was conducted in accordance with the SPE-LC-MS/MS protocol “BfR-PA-Tea-2.0/2014” [74] published by the BfR (Bundesinstitut für Risikobewertung, Berlin, Germany). The analytical method comprises 28 reference PA compounds (echimidine, echimidine-*N*-oxide, erucifoline, erucifoline-*N*-oxide, europine, europine-*N*-oxide, heliotrine, heliotrine-*N*-oxide, intermedine, intermedine-*N*-oxide, jacobine, jacobine-*N*-oxide, lasiocarpine, lasiocarpine-*N*-oxide, lycopsamine, lycopsamine-*N*-oxide, monocrotaline, monocrotaline-*N*-oxide, retrorsine, retrorsine-*N*-oxide, senecionine, senecionine-*N*-oxide, seneciphylline, seneciphylline-*N*-oxide, senecivernine, senecivernine-*N*-oxide, senkirkine, trichodesmine). From these PAs analyzed, only nine were detectable in our *S. vulgaris* samples. The statistical analysis of the data was performed using IBM^®^ SPSS^®^ Statistics 24.

## Figures and Tables

**Figure 1 plants-08-00054-f001:**
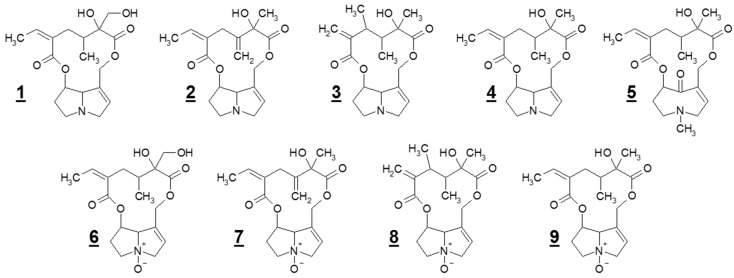
Chemical structures of the nine pyrrolizidine alkaloids (PAs) detected in *Senecio vulgaris* L. in this study: (**1**) retrorsine (Re); (**2**) seneciphylline (Sp); (**3**) senecivernine (Sv); (**4**) senecionine (Sc); (**5**) senkirkine (Sk, otonecine derivative); (**6**) retrorsine-*N*-oxide (ReN); (**7**) seneciphylline-*N*-oxide (SpN); (**8**) senecivernine-*N*-oxide (SvN); and (**9**) senecionine-*N*-oxide (ScN).

**Figure 2 plants-08-00054-f002:**
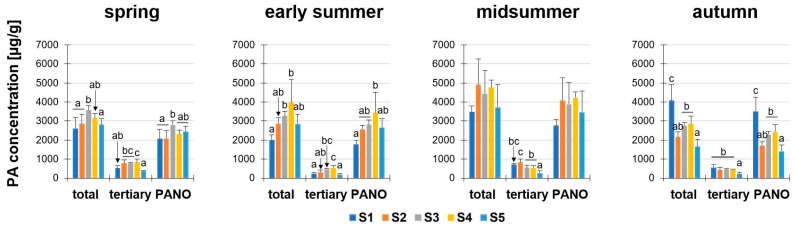
PA concentration µg/g in dry matter (dm) of *Senecio vulgaris* L. plants depending on the developmental stage (S1 to S5) and season. Data are values for (i) total concentration of PAs (“total”; tertiary PAs and PA-*N*-oxides), (ii) concentration of tertiary PAs (“tertiary”; Re, Sp, Sv, Sc, Sk) and (iii) concentration of PA-*N*-oxides (“PANO”; ReN, SpN, SvN, ScN). Results are means ± SD of five different determinations (distinct sub-trials, see Appendix A). Different letters identify significant differences (*p* < 0.05) between developmental stages (ANOVA, Tukey’s honest significance difference).

**Figure 3 plants-08-00054-f003:**
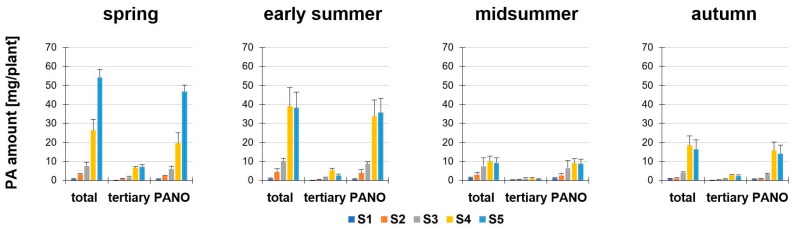
PA amount of *Senecio vulgaris* L. (mg/plant) depending on the developmental stage (S1 to S5) and season. Data are values for (i) total amount of PAs (“total”; tertiary PAs and PA-*N*-oxides), (ii) amount of tertiary PAs (“tertiary”; Re, Sp, Sv, Sc, Sk) and (iii) amount of PA-*N*-oxides (“PANO”; ReN, SpN, SvN, ScN). Results are means ± SD of five different determinations (distinct sub-trials, see Appendix A).

**Figure 4 plants-08-00054-f004:**
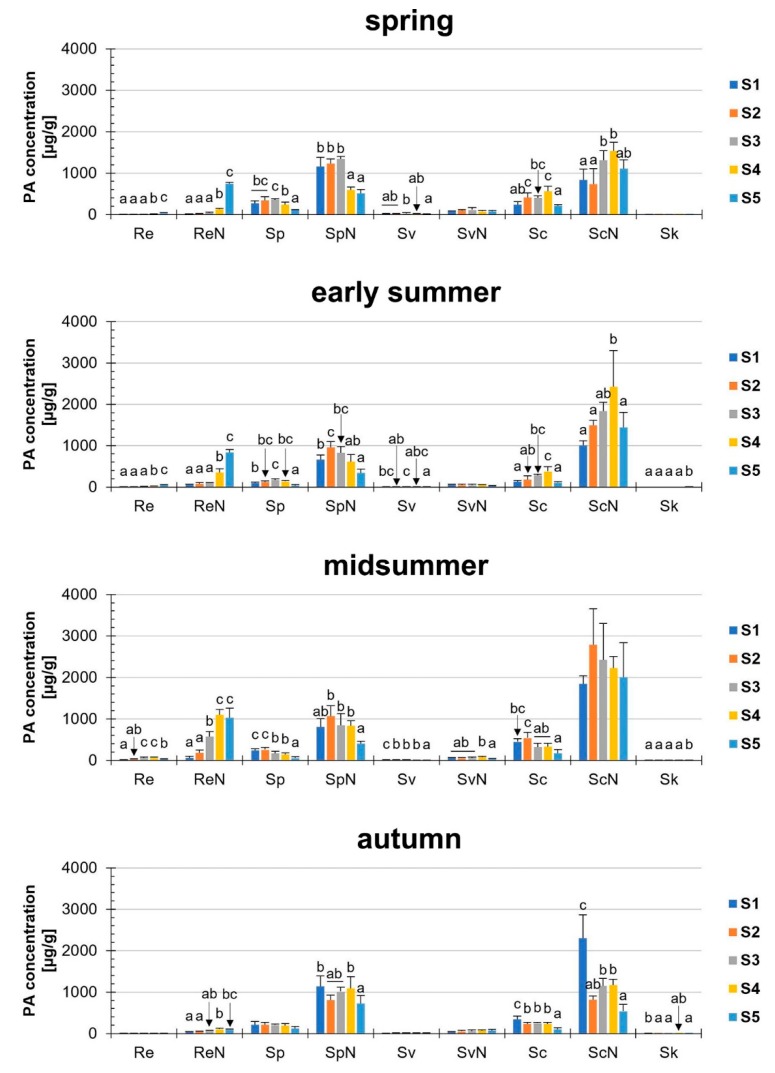
PA concentration (µg/g dm) of nine different PAs in *Senecio vulgaris* L. plants depending on the developmental stage (S1 to S5) and season. Results are means ± SD of five different determinations (distinct sub-trials, see Appendix A). Different letters identify significant differences (*p* < 0.05) between developmental stages (ANOVA, Tukey’s honest significance difference). Data for Re, Sv, Sk and SvN are shown in a different scaling in Appendix A.

**Table 1 plants-08-00054-t001:** Sampling dates for the distinct developmental stages and seasons.

Season	Developmental Stage				
	S-1	S-2	S-3	S-4	S-5
**spring**	16 April	25 April	4 May	16 May	31 May
**early summer**	2 June	12 June	21 June	30 June	14 July
**midsummer**	20 July	27 July	4 August	13 August	25 August
**autumn**	15 September	23 September	5 October	3 November	3 December

**Table 2 plants-08-00054-t002:** Definition and description of the stages of development for *Senecio vulgaris* L. to be sampled, based on the BBCH monograph of Meier [73].

Developmental Stage	Description	Based on BBCH Code (Micro Stage)
**S1**	7 to 9, or more true leaves unfolded	17–19
**S2**	3 to 6 visible sprouts	23–26
**S3**	first individual flowers visible (still closed) until the beginning of flowering (10% of flowers open)	55–61
**S4**	full flowering (50% of flowers open)	65
**S5**	fruit development until fully ripe seeds	72–89

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
