# Peer review of "Occurrence of Nine Pyrrolizidine Alkaloids in Senecio vulgaris L. Depending on Developmental Stage and Season"

_plants, 2019, doi:10.3390/plants8030054_

Round 1

Reviewer 1 Report

The manuscript "Occurrence of nine pyrrolizidine alkaloids in Senecio vulgaris L. depending on developmental stage and season" deals with the detection of 9 PA in dependence of the season and the developmental stage of Senecio vulgaris.

I have only a few comments to adress:

Introduction:

-Please change the first sentence of the introduction, it is difficult to understand due to its length.
-Figure caption Fig 1: Please indicate clearly, that you also measured the PANO

-What was the reason for only analyzig this 9 PA? Do you check for addditional PA (heliotridine, echimidine, lasiocarpine,...) as you mentioned in the Material and methods section, the method you used comprise 13 PA, you measured only 9 PA

-p80/line 72: please use elucidate instead of find out

Results:

- general remark: in my opinion: "concentration" of a compound is referred to a volume (defined DIN1310): you used concentration for describin µg/g plant...this is not correct...please revise the whole manuscript and the figures

-in my opinion, it would be helpful for the manuscript, if in the results section the different stages of development are explained shortly

-statistics: it is unclear what is the reference point for your analysis, and what is the meaning of the letters, please indicate in each figure legend the reference point for statistics, please revise all figure captions!

-table 1: table caption: I my opinion significance can not be increased, please revise

For me, the content of this table is not clear. What is the reference point for statistical analysis and the main take home message of this table??? Without any content the content is not necessary (data are presented well in figure 4).

Discussion:

-Please say a few words about normal human exposure. Normally, people ingest PA with food as a contaminant. Please integrate some few date  PA food contents in Europe, not only about herbal medicines.

Author Response

ANSWER TO REVIEWER

The manuscript "Occurrence of nine pyrrolizidine alkaloids in Senecio vulgaris L. depending on developmental stage and season" deals with the detection of 9 PA in dependence of the season and the developmental stage of Senecio vulgaris.

We would like to thank the reviewer for his/her time in reading our manuscript and for the encouraging comments. His/her comments and suggestions have helped us improve the quality of our manuscript and are greatly appreciated.

I have only a few comments to address:

Introduction:

-Please change the first sentence of the introduction, it is difficult to understand due to its length. 

Thank you for the comment, we changed the first sentence according your suggestions.

“Pyrrolizidine alkaloids (PAs) are phytochemicals occurring in more than 6,000 plant species [1-2], e.g. Asteraceae, Boraginaceae, Fabaceae and Orchidaceae families [3-4]. Further 700 plant species are indicated to be able to form PAs [5].”

-Figure caption Fig 1: Please indicate clearly, that you also measured the PANO

Thank you very much for your suggestion. We included the structures of the N-oxides in the figure 1 and changed the figure legend as follows:

“Chemical structures of the nine PAs detected in Senecio vulgaris L. in this study: 1: retrorsine (Re), 2: seneciphylline (Sp), 3: senecivernine (Sv), 4: senecionine (Sc), 5: senkirkine (Sk, otonecine derivative), 6: retrorsine-N-oxide (ReN), 7: seneciphylline-N-oxide (SpN), 8: senecivernine-N-oxide (SvN) and 9: senecionine-N-oxide (ScN)”

-What was the reason for only analyzig this 9 PA? Do you check for addditional PA (heliotridine, echimidine, lasiocarpine,...) as you mentioned in the Material and methods section, the method you used comprise 13 PA, you measured only 9 PA

Thank you very much for this question. This point was not correctly addressed in the manuscript. We analyzed all 28 compounds described in the analytical method established by the BfR [60]. Only 9 PAs were detectable in our samples.

Since the corresponding sentence in the material and methods part is somewhat misleading, we changed it as follows:

“The analytical method comprises 28 reference PA compounds (senecionine, senecionine-N-oxide, seneciphylline, seneciphylline-N-oxide, monocrotaline, monocrotaline-N-oxide, retrorsine, heliotrine, heliotrine-N-oxide, trichodesmine, retrorsine-N-oxide, echimidine, intermedine, lycopsamine, senkirkine, lasiocarpine, lasiocarpine-N-oxide, europine-N-oxide, europinehydrochloride, echimidine-N-oxide, erucifoline, erucifoline-N-oxide, intermedine-N-oxide, jacobine, jacobine-N-oxide, lycopsamine-N-oxide, senecivernine, senecivernine-N-oxide). From these PAs analyzed, only nine were detectable in our S. vulgaris samples.”

-p80/line 72: please use elucidate instead of find out

This was changed according to your suggestion

Results:

- general remark: in my opinion: "concentration" of a compound is referred to a volume (defined DIN1310): you used concentration for describin µg/g plant...this is not correct...please revise the whole manuscript and the figures

Thank you for your comment! We agree that the term „concentration“ is not used according DIN1310 in our manuscript. However within the research field of plant nutrition, it is commonly used in this way (concentration: µg/g dry mass; content: µg/plant. Here we show some recent publications concerning PA “concentrations” in plants using a terminology which is not congruent with DIN1310

Picron et al. Food Chem. 2018 Nov 15;266:514-523; “Total PAs concentration of samples purchased on the Belgian market varied greatly with matrix type ranging from <LOD to 187151 ng/g for dry samples”

Letsyo et al. Phytother Res. 2017 Dec;31(12):1903-1909; “the average PA concentration of the samples was 201 μg/kg, the highest concentration of PAs (3270 μg/kg) was attributed to a product that was purchased…”

Kast et al. Food Addit Contam Part A Chem Anal Control Expo Risk Assess. 2018 Feb;35(2):316-327. “Of the commercially available pollen, 31% contained PAs with a mean concentration of 319 ng/g, mainly”

-in my opinion, it would be helpful for the manuscript, if in the results section the different stages of development are explained shortly

Thank you for your comment! We agree that it is important to describe the different stages of development adequately. Additional to the table 3 describing the most important differences between the stages, we added a note to this table in the first sentence: “In four seasons (spring, early summer, midsummer, autumn) nine PAs were investigated in five developmental stages (S1 to S5, according to the description in table 3).”

-statistics: it is unclear what is the reference point for your analysis, and what is the meaning of the letters, please indicate in each figure legend the reference point for statistics, please revise all figure captions!

Thank you for your comment. You are right that no “reference point” is shown in the figure legend. The correlation with a specific reference point was not intended, since we just showed which values differ with a statistical significance of <0.05. So every value in a group is compared with all other values in the group. Due to the multiple comparisms, the result of this analysis cannot be shown using asterisks, but using letters: This was stated clearly in the figure legend:

 “Different letters identify significant differences (p<0.05) between developmental stages (ANOVA, Tukey-HSD)”

-table 1: table caption: I my opinion significance can not be increased, please revise

For me, the content of this table is not clear. What is the reference point for statistical analysis and the main take home message of this table??? Without any content the content is not necessary (data are presented well in figure 4). 

Thank you for your comment, we deleted table 1 according to your suggestions.

Discussion:

-Please say a few words about normal human exposure. Normally, people ingest PA with food as a contaminant. Please integrate some few date  PA food contents in Europe, not only about herbal medicines. 

Thank you for your suggestion. We added the following sentence in the introduction:

According to calculations of the German Federal Institute for Risk Assessment (BfR), the intake of 1,2-unsaturated PAs by children aged six months to five years is mainly attributable to herbal teas (incl. rooibos tea), black tea and honey. With adults, the contribution to total 1,2-unsaturated PA intake made by honey is lower, and that of green tea higher, than with children [10, 50, 60]. As a consequence, contamination of herbal products with S. vulgaris may be a relevant source of PAs.

Reviewer 2 Report

See attached file.

Author Response

Reviewer’s comments

Journal: Plants

Manuscript ID: 437926

General comments:

I understand that it is important to minimize the risk of PA intoxication, and that, therefore, the levels of the various PAs should be determined. However, in the case of plants for nutritional and medicinal purposes, which are contaminated by S. vulgaris (line 77-78), I miss an additional comment about the rules and regulations of this type of contamination. Are producers obliged to check whether their samples are contaminated with S. vulgaris (or other PA containing plants)? What are the accepted limits with regard to this?

Thank you for your comment. We added the following sentences:

“At the moment there are no legal limit values for 1,2-unsaturated PAs in foods and feeds. Within the European Union, the general recommendation applies that exposure to genotoxic and carcinogenic substances should be minimised to the lowest level achievable by reasonable means (ALARA principle: as low as reasonably achievable) [50 – 52].”

In the results of the total PA concentration in dry matter of S. vulgaris, I see large standard deviations. The averages are found after LC-MS analysis of 5 different samples. It would be nice to check the variation in results of the LC-MS analysis alone and of the same sample extracted multiple times, to confirm that the majority of the variation in your results originates from variation in the samples and not from the handling of the samples.

Thank you for your comment. Since the LC-MS analysis was performed by the Institut Kirchhoff Berlin GmbH, a well-established laboratory for analytical question with all accreditations and authorisations for detection of e.g. food samples (accreditation to DIN EN ISO/IEC 17025 since 1993, and notified to European Regulation (EC) 882/2004 since 2000), we suggest that the variations are caused by the samples.

Further information about the lab: https://www.institut-kirchhoff.de/quality-management/quality-assurance/?L=1

Specific comments:

- line 37/38, line 48: family names should not be written in italics.

Thank you for your comment. We changed this in the manuscript

- line 42: in consequences of

Thank you for your comment. We changed this in the manuscript

- figure 1: not necessary to give abbreviations of the compounds? Instead, I would mention those in the text.

Thank you for your comment: We changed the figure 1 including the structures of the corresponding N-oxides and additionally changed the figure legend.

- line 62: Senecio

Thank you for your comment. We changed this in the manuscript

- line 66: disbalance

Thank you for your comment. We changed this in the manuscript

- line 67-69: I think the risks to humans and animals are known, but can fluctuate from batch to batch, depending from the level of PAs, so I would suggest to rewrite this sentence.

Thank you for your comment. We changed the sentence according to your suggestions

“Due to the high variability of PAs in different plants (PA amount, relationship of PAs vs. PANOs, occurrence of distinct PA structures…), a general estimation of the toxicological risk of PA containing plants for humans is rather problematic.”

- line 70, 71: “influence the PA concentrations and amounts as well as causing the variability of PAs in S. vulgaris” the first part and second part of the sentence have the same meaning, so one part can be omitted. Moreover, the sentence is grammatically not correct.

Thank you for your comment. We changed this in the manuscript as follows:

We used HPLC-MS analysis to clarify if the concentration and amount of the total PAs in S. vulgaris is influenced by ontogenesis and season. Furthermore, we investigated the influence of ontogenesis and season on the nine distinct PAs which were detected in this plant.

- line 86: S. vulgaris in italics

We changed this in the manuscript

- line 90-92: “low standard deviations” When I calculate the RSD, I don’t think you can say that your standard deviations can be considered as low? I don’t understand which discrepancy was significant.

Thank you for your comment! You are right, the SD values are not very low. We don’t believe that there is a difference in the total PA concentration in spring. However, the analysis of statistical significance detects a significance in the amount of total PAs in spring between S1 and S3 which was shown by different letters in the figure (a, b). Since we don’t believe that this difference is really relevant, we wrote as a second part of the sentence: “…but probably not relevant for biological systems.”.

We changed the sentence to: “This discrepancy is significant, but probably not relevant for biological systems.”

- figure 2 and figure 4 and supplemental figure S4: on the y-axes, all numbers should be written with a comma, not with a full stop! Now, for example in figure 2, the axes range is from 0 to 7, but what you meant is a scale from 0 to 7000!

Thank you for your comment! We changed figure 2, 4, S4  and S5 according to your comment.

- line 116-117: a significant difference in all seasons was observed

We changed this in the manuscript

- line 121: “42 and 110 %”, “230% and 130%”. What do these values mean exactly?

We changed this sentence according to your suggestions:

“PA levels fell in the autumn in S4 and S5, compared to spring (about 42 % and 230 %) and early summer (about 110 % and 130 %).”

- line 137: “all alkaloids were affected ontogenetically at least three times during the course of the year” this seems like a strange comment to me and I don’t know what you mean exactly by “affected at least three times”.

Thank you for your comment, we deleted this sentence from the manuscript

- table 1: here I would mention the full names of the compounds

According to the comments of the other reviewers we had to delete table 1 from the manuscript.

- line 150: very small concentrations  very low concentrations

We changed this in the manuscript

- line 152: “the amount of was up to: there is a compound name missing.

We changed this in the manuscript to:

“Regarding the corresponding N-oxide (ReN) in midsummer, S4; the concentration of ReN was approximately eighteen-fold higher (1099.3 µg/g) than that of Re.”

- line 212-217: the concentration of PAs declines in autumn. You state that the fact that the dry matter was also lower in autumn, causes an increased concentration of the PAs, however, this is in contrast with your finding of the reduced levels?

Thank you for your comment. We explain this finding by a decline of PAs synthesis in the plant.

- line 233: may undergo

We changed this in the manuscript

- line 271: 60 °C

We changed this in the manuscript, as also in other 3 temperature values in materials an methods

- line 309: 3.8 cm

We changed this in the manuscript

- line 309: 21 mL

We changed this in the manuscript

- line 310 and 311: “S. vulgaris” is enough

We changed this in the manuscript. Thank you, for your comment! According to Supplemental Figure S3 and S5, we decide to maintain our description

- line 335: this is supplemental figure S6, not S4

We changed this in the manuscript … according to that we detailed the figure capitation in a correct way : “PA concentration [µg/g] of A: retrorsine; B: senecivernine; C: senkirkine; D: senecivernine-N-oxide in Senecio vulgaris L. plants (dry mass) depending on the developmental stage (S1 to S5) and season.”

Reviewer 3 Report

Review nine PAs Senecio vulgaris

The present manuscript is devoted to the investigation of nine pyrrolizine alkaloids (PAs) in Senecio vulgaris at different development stages of the plant at different seasons. As S. vulgaris is considered one of the  main contamination sources of other herbs used in phytopharmaceuticals and herbal teas, this manuscript provides interesting insight into the concentrations of PAs at different plant stages. It was found that the N-oxides of PAs (PANO) predominate through all development stages at all seasons and that the number of total PAs doesn’t change significantly. Although previous studies hava also determined the total PAs in S. vulgaris, this study provides more detailed insight into the content of each of the nine PAs and how it changes with time. Therefore this paper is worth publishing in plants after considering the following minor revision:

1.     Introduction

The references should be completed by the studies devoted to PAs in medicinal herbal teas and honey e.g the following studies:

Detection of pyrrolizidine alkaloids in German licensed medicinal herbal teas

or

determination of pyrrolizidine alkaloids in tea, herbal drugs and honey.

2.     Results

Page 3 line 83: reference to table 3 should be made here, so that the reader can clearly understand the different stages of develoment

Page 3 line 99: the Authors mention a ratio of tertiary PAs to PANOs ranging from 10-30% to 70-90%. However,  the calimed ration of 70-90% is not obvious from the figures, as tertiary PAs are always much smaller in amount compared to the PANOs.

Figure 2: the amounts of PAs determined in the midsummer are characterized by very high standard deviation. Do you have an idea for this higher standard deviations compared to the other seasons?

Page 4 line 146: Supplemental Figure 6 is missing.

3.     Discussion

Page 7 line 242: the different toxic potential of the different PAs should be discussed in more details.

And finally a last question:

Why didn’t the important PA in Senecio erucifoline was not included in the study? In this context it should be explained, why especially the present nine alkaloids were chosen and no others.

Author Response

The present manuscript is devoted to the investigation of nine pyrrolizine alkaloids (PAs) in Senecio vulgaris at different development stages of the plant at different seasons. As S. vulgaris is considered one of the  main contamination sources of other herbs used in phytopharmaceuticals and herbal teas, this manuscript provides interesting insight into the concentrations of PAs at different plant stages. It was found that the N-oxides of PAs (PANO) predominate through all development stages at all seasons and that the number of total PAs doesn’t change significantly. Although previous studies hava also determined the total PAs in S. vulgaris, this study provides more detailed insight into the content of each of the nine PAs and how it changes with time. Therefore this paper is worth publishing in plants after considering the following minor revision:

We would like to thank the reviewer for his/her time in reading our manuscript and for the encouraging comments. His/her understanding of the significance of our work is greatly appreciated.

1.     Introduction

The references should be completed by the studies devoted to PAs in medicinal herbal teas and honey e.g the following studies:

Detection of pyrrolizidine alkaloids in German licensed medicinal herbal teas

 or

 determination of pyrrolizidine alkaloids in tea, herbal drugs and honey.

Thank you for your comment. We added the following part in the introduction section:

“At the moment there are no legal limit values for 1,2-unsaturated PAs in foods and feeds. Within the European Union, the general recommendation applies that exposure to genotoxic and carcinogenic substances should be minimised to the lowest level achievable by reasonable means (ALARA principle: as low as reasonably achievable) [50 52]. From official surveys, experimental work and reviews carried out or compiled for herbal products and honey, it becomes clear that a wide range of matrices consumed can be contaminated with PAs: (i) honey and pollen [11; 21; 53 59], (ii) herbal medicines [51; 60 61], (iii) herbal supplements and foods [9; 11; 13; 15; 20; 59] and (iiii) tea and herbal teas [60 66]. According to calculations of the German Federal Institute for Risk Assessment (BfR), the intake of 1,2-unsaturated PAs by children aged six months to five years is mainly attributable to herbal teas (incl. rooibos tea), black tea and honey. With adults, the contribution to total 1,2-unsaturated PA intake made by honey is lower, and that of green tea higher, than with children [10, 50]. As a consequence, contamination of herbal products with S. vulgaris may be a relevant source of PAs.”

Accordingly, the bibliography has increased and reference numbers have been assigned new.

2.     Results

 Page 3 line 83: reference to table 3 should be made here, so that the reader can clearly understand the different stages of develoment

Thank you for your comment! We agree that it is important to describe the different stages of development adequately. The sentence in line 83 was changed to:

“In four seasons (spring, early summer, midsummer, autumn) nine PAs were investigated in five developmental stages (S1 to S5, according to the description in table 3).”

Page 3 line 99: the Authors mention a ratio of tertiary PAs to PANOs ranging from 10-30% to 70-90%. However,  the calimed ration of 70-90% is not obvious from the figures, as tertiary PAs are always much smaller in amount compared to the PANOs.

Thank you for your comment! We changed the sentence according to your suggestions:

Consistently in all developmental stages and in each season the concentrations of the PA-N-oxides (PANOs) were higher than the tertiary PAs in the plant, leading to ratios of tertiary PAs to PANOs between 10:90 % to 30:70 %.”

Figure 2: the amounts of PAs determined in the midsummer are characterized by very high standard deviation. Do you have an idea for this higher standard deviations compared to the other seasons?

Thank you for your comment. In midsummer, S. vulgaris showed no significant differences in the concentration of the total PAs due to the large variance of the values in the developmental stages S2, S3 and S5. This is explained by the weather conditions in midsummer (very below average precipitation levels in July and August 2016), which may have caused a stagnation or decrease in fresh weight and only a small increase in dry matter. The deteriorated growth of the plants also explains why the variances of dry matter in the summer were very high, thereby affecting the levels in concentration and the amount of the PAs in the plants.

Page 4 line 146: Supplemental Figure 6 is missing.

 Thank you very much for your comment. Supplemental Figure 6 was unintendently named Supplemental Figure 4. We changed this in the manuscript. 

3.     Discussion

Page 7 line 242: the different toxic potential of the different PAs should be discussed in more details.

Thank you very much for your comment. We added the following part in the manuscript:

In animal studies, certain PAs found in S. vulgaris show hepatoxic as well as carcinogenic and genotoxic effects. This applies to PAs with a 1,2-unsaturated necine base that is further esterified with at least one branched C5-carboxylic acid. The corresponding PA-N-oxides are less toxic [10, 50, 60].

And finally a last question:

Why didn’t the important PA in Senecio erucifoline was not included in the study? In this context it should be explained, why especially the present nine alkaloids were chosen and no others.

Thank you very much for this comment. This point was not correctly addressed in the manuscript. We analyzed all 28 compounds described in the analytical method established by the BfR [60] including erucifoline, which was not detectable.

Since the sentence in the material and methods part is somewhat misleading, we changed it as follows:

““The analytical method comprises 28 reference PA compounds (senecionine, senecionine-N-oxide, seneciphylline, seneciphylline-N-oxide, monocrotaline, monocrotaline-N-oxide, retrorsine, heliotrine, heliotrine-N-oxide, trichodesmine, retrorsine-N-oxide, echimidine, intermedine, lycopsamine, senkirkine, lasiocarpine, lasiocarpine-N-oxide, europine-N-oxide, europinehydrochloride, echimidine-N-oxide, erucifoline, erucifoline-N-oxide, intermedine-N-oxide, jacobine, jacobine-N-oxide, lycopsamine-N-oxide, senecivernine, senecivernine-N-oxide). From these PAs analyzed, only nine were detectable in our S. vulgaris samples.”
